# Anticonvulsant effects of novel and repurposed drugs on docetaxel-induced neuropathy in *C. elegans*

**Paola Ximena Gonzalez-Lerma**[1,2,3¤a], **Crystal Lloyd**[3,4], **Scarlet J. Park**[3¤b],
**Ken Dawson-Scully**[1,3,5¤c]*

1 Department of Integrative Biology, Charles E. Schmidt College of Science, Florida Atlantic University, Boca Raton, Florida, United States of America, 2 Department of Biomedical Sciences, Charles E. Schmidt College of Medicine, Florida Atlantic University, Boca Raton, Florida, United States of America, 3 Division of Research and Economic Development, Nova Southeastern University, Davie, Florida, United States of America, 4 College of Pharmacy, Nova Southeastern University, Palm Beach Gardens, Florida, United States of America, 5 Department of Psychology and Neuroscience, College of Psychology, Nova Southeastern University, Davie, Florida, United States of America

¤a Current Address: Allosite Therapeutics, Miami, Florida, United States of America
¤b Current Address: Department of Research and Economic Development, Nova Southeastern University, Davie, Florida, United States of America
¤c Current Address: Office of the Provost and Executive Vice President for Academic Affairs, Florida Atlantic University, Boca Raton, Florida, United States of America
* kds@fau.edu

## Abstract

Chemotherapeutic agents used for most common cancers are frequently associated with neurotoxicity, which often include debilitating side effects such as seizures. Docetaxel, one of the most widely and effectively used chemotherapeutic drugs, is associated with an array of symptoms referred to as Docetaxel-Induced Peripheral Neuropathies (DIPNs), including motor neuropathy, tingling, muscle weakness, and numbness. In this study, we use the electroconvulsive assay to model DIPN-related muscle weakness in *C. elegans,* via shock induction. We show that acutely or chronically exposing nematodes to docetaxel increases time to recovery from shock-induced seizure-like behaviors. Additionally, we find that sildenafil citrate, a PDE-5 inhibitor, and a novel bicyclic bridge compound, Resveramorph-3 (RVM-3), are both effective at rescuing the animals from prolonged seizure-like movement duration following acute and chronic exposure to docetaxel. The results demonstrate that sildenafil citrate and RVM-3 are potential candidates for mitigating the neurological deficits resulting from DIPNs.

## Introduction

Although cancer remains a leading cause of death globally, significant strides in its diagnosis and treatment have dramatically increased survival rates. For example, 69% of cancer patients in the U.S. receive a 5+year survival prognosis [1].

**Data availability statement:** All datasets generated and/or analyzed during the current study are available in Open Science Framework DOI: 10.17605/OSF.IO/S56DM.

**Funding:** The author(s) received no specific funding for this work.

**Competing interests:** The authors have declared that no competing interests exist.

Consequently, addressing the long-term toxicity of cancer treatments is increasingly critical for quality of life [2].

Chemotherapy treatments, while effective against cancer, often result in significant long-term toxicities. Among these, Chemotherapy-Induced Peripheral Neuropathies (CIPNs) are particularly debilitating, affecting the central and peripheral nervous systems and impacting up to 85% of the patients and survivors of cancer [3]. Docetaxel is a taxane which disrupts microtubule-mediated cell division and the dynamic assembly of polymer microtubule subunits, triggering peripheral neuropathies through damaged nerve terminals, neuronal axons, and cellular mitochondria [4]. Intravenously administered docetaxel has been shown to be effective for treating advanced, metastatic, or chemotherapy-resistant cancers, such as androgen-independent prostate cancer and head and neck squamous carcinoma, but its neurotoxic side effects often lead patients to abandon treatment [5,6].

Studies have shown that several drugs used in the treatment of cancer, can trigger seizures [7]. Furthermore, there have been reported cases of docetaxel infusion leading to dose-dependent peripheral sensory [8] or motor [9] neuropathy in cancer patients. Exposure to taxanes, which includes docetaxel, can result in reduced motor and sensory nerve action potentials, along with decreased motor nerve conduction velocity [10–12]. Other studies in breast cancer patients treated with docetaxel reported motor peripheral neuropathy as more common than sensory peripheral neuropathy [13]. In this work we focus on the effects that docetaxel has in modulating time to recovery from shock-induced seizure-like behaviors in treated *Caenorhabditis elegans*.

The genetically tractable nematodes have a mapped neuronal circuitry, which facilitates identification of neuronal mechanisms that correlate with locomotive phenotypes [14–16]. Previous works have established *C. elegans* as a useful model for studying the cellular and molecular basis for structural changes in axons [17] that accompany peripheral neuropathy, particularly including following exposure to paclitaxel, a taxane [17]. As taxanes mainly disrupt microtubule-mediated transport and axonal integrity, we hypothesized that docetaxel, which is another taxane, may induce similar mechano-toxic effects observed following paclitaxel treatment in nematodes.

Our lab has previously used electroshock to model seizure-like behaviors in *C. elegans*, including drug exposure [18–21]. In this study, we use *C. elegans* to establish a model of shock-induced seizure-like behaviors and evaluate drug-induced changes in recovery time for the purpose of identifying potential rescue agents. We show that acute and chronic exposure to docetaxel increases the duration and severity of shock-induced seizure-like behaviors in worms. Furthermore, these docetaxel-induced effects were alleviated by concurrent administration of sildenafil citrate, an indirect protein kinase G (PKG) activator [20], and Resveramorph-3, a novel bicyclic compound [22]. The results from this study identify potential candidates for mitigating the neurological deficits resulting from DIPNs.

## Materials and methods

### *C. elegans* stocks and cultivation

All animals used in this study were *C. elegans* of the N2 strain, a commonly used control strain. The worms were acquired from the Caenorhabditis Genetics Center

(CGC) at the University of Minnesota. Stocks were maintained and transferred every 3–4 days on standard Nematode Growth Medium (NGM) agar plates seeded with OP50 *E. coli.* The animals were reared and maintained in a temperature-controlled setting (20 °C, except the day before the experiment). Worms were transferred using a platinum wire pick, sterilized between transfers by a butane flame.

## Acute exposure protocol

On day 1, adult worms with eggs were plated on NGM agar plates seeded with OP50 *E. coli* for 2 days at 20 °C. On day 3, L4-stage worms were picked and plated on a new NGM agar plate seeded with OP50 *E. coli* for overnight incubation at 22 °C. On day 4, 1-day-old adult worms were incubated for 30 minutes in M9 saline (control) or in the respective test solution prior to the electroshock delivery. Docetaxel and sildenafil citrate were first dissolved in 1% dimethyl sulfoxide (DMSO), followed by a 1:10 dilution in M9 saline. Therefore, the tested solutions were dissolved in 0.1% DMSO of the total volume with a 99.9% M9 saline. RVM-3 was dissolved directly in M9 saline.

## Chronic exposure protocol

For chronic exposure, *E. coli*-seeded NGM agar plates were coated with M9 saline containing the specified concentration of docetaxel or vehicle control (M9 with 0.1% v/v DMSO; see next section). In detail, docetaxel was first dissolved in 1% DMSO, followed by a 1:10 dilution in M9 saline (this is equivalent to the acute experiments). Therefore, the tested solutions were dissolved in 0.1% DMSO of the total volume with a 99.9% M9 saline. As for plate preparation, 10 milliliters of NGM agar were first dispensed per plate and once the agar solidified, 2 milliliters of corresponding solutions were poured on the surface of the agar plates. Therefore, for chronic concentrations, DMSO was further dissolved in a total of 12 milliliters, resulting in a final DMSO concentration of ~0.01%, rather than the 0.1% used for acute experiments. The final drug concentrations used for chronic experiments take into account the additional volumes. Plates were then covered and left to dry for 24 hours, away from light at room temperature, to avoid light and temperature degradation. All plates used in this protocol were standardized. After plates air-dried with the coated layer of corresponding solution, these were ready to use for experiments. Ten plates were prepared per condition. New batches of plates were prepared every week to avoid drug degradation through time. Using the electroshock assay protocol, we tested each chronic plate per condition against the acute protocol to ensure compounds distributed evenly and retained bioactivity on the agar. For experiments, nematodes were placed on the surface of each plate. It is important to note that *C. elegans* do not burrow into the agar, but rather swim on the surface of the agar. On day 1, six gravid adult worms were transferred onto the surface of each coated plate and incubated for 2 days at 20 °C. On day 3, L4-stage worms were transferred to a new drug-treated plate for overnight incubation at 22 °C. On day 4, 1-day-old adult worms were incubated for 30 minutes in M9 saline (control) or the test solution, prior to the electroshock delivery.

## Drugs and chemicals used

Drugs and solutions used in the electroconvulsive shock assay were: M9 saline (0.022M $KH_2PO_4$, 0.042M $Na_2HPO_4$, 0.085M NaCl, 0.008M $MgSO_4$), docetaxel (ThermoFisher Scientific, CAS: 114977-26-5), and sildenafil citrate (Viagra®; Alabama Pharmacy Industry Solutions, CAS: 171599-83-0). Resveramorph-3 (RVM-3) was synthesized by the Lepore Lab at Florida Atlantic University. Docetaxel and sildenafil citrate were dissolved first in 1% DMSO and subsequently diluted 1:10 in M9 saline to achieve the final desired drug concentrations. This means that tested solutions contained 0.1% DMSO and 99.9% M9 saline. DMSO alone could have neuroprotective properties when employed at low concentrations. However, previously in our lab a DMSO dose curve was completed showing that there is no significant difference in recovery from electric shock in concentrations up to 0.5% DMSO [19]. RVM-3 is a novel compound previously synthesized and studied for solubility by our lab [22]. RVM-3 has a molecular weight of 418.44 g/mol. For this study, lyophilized RVM-3

was stored in parafilm sealed amber glass vials away from light at 20 °C for a period of up to three months. The day of experiments, 1 milligram of lyophilized RVM-3 was resuspended directly into M9 saline to make a stock concentration of 100 μM. RVM-3 solutions were stored in parafilm sealed amber glass vials away from light at 20 °C for a period of up to two weeks.

## Electroshock assay

*C. elegans* possess 302 neurons, conserved neurotransmitter systems, a simplified neuronal circuitry, and a fully mapped connectome, making it a powerful model for neurological studies. Building on our previous work [18–21], we established an electroshock assay for pharmacological screening. This assay elicits a shock-induced seizure-like behavior that serves as a behavioral indicator of locomotor circuitry and neuronal function, enabling the identification of compounds that influence recovery time from these behaviors. The electroshock assay was conducted as previously published [18–21] with minor modifications. For the electroconvulsive shock assay, the setup consisted of a Grass SD9 stimulator, Grass SD44 stimulator, a dissecting stereoscope (AmScope SM-1TSX) with a camera (HY-1139), and a computer with OBS Studio recording software. Briefly, approximately six 1-day-old adult *C. elegans* were transferred into silicone tubes. Following 30 minutes of incubation, two 18-gauge copper wires were inserted into either end of the tube to a 1-cm distance between the electrodes. Alligator clips were attached to each electrode and connected to the stimulator. Worms were monitored for 30 seconds before and 5 minutes after a 3-sec shock delivery (200 Hz, 47 V). New electrodes were used for each test solution, and the experimental tubes were discarded after each trial. The shock delivery was visually confirmed as electrolysis in the form of bubbles released from either electrode. Video recordings were analyzed for the recovery time of each worm after the shock-induced seizure-like behaviors. Nematodes occluded from view or those that did not display normal movement before and after shock were excluded from analysis. Animals that recovered within the five-minute window of recording were counted as recovered. Recovery time of a worm was defined as time at which the nematode initiated three consecutive sinusoidal wave-like swimming motions, without considering the speed or depth of the wave. Animals that were active prior to shock but did not recover following shock, were considered as non-recovered (NR) and were used to calculate percent non-recovery (%NR) for each solution tested using the following formula: $\%NR = \frac{(\text{Number of non–recovered worms})}{(\text{Total number of worms analyzed})} \times 100$. For accurate screening, we ran six tubes per condition, each tube containing six nematodes. To avoid day effects on our screening, we ran two tubes per condition per day. For clarification, each day we included a trial of M9 saline as a negative control. To maintain drug screening and video analysis objectivity, two researchers were involved in collecting data for this study. Each researcher was responsible for completing randomly assigned dosage curves on different days of the week as mentioned previously. To eliminate personal bias during the video analysis process, videos were swapped between the two researchers and scores were blinded to condition.

## Statistical methods

For recovery time, one-way ANOVA was followed by all pairwise Student-Newman-Keuls post hoc test. Chi-square tests were used to compare % NR. Detailed statistical results can be found in the supplemental material. An alpha value of 0.05 to determine significance. All ANOVA statistics were performed using SigmaPlot (Version 13) and Chi-Square Tests were performed using the Standard Deviation Chi-Square Calculator.

## Results

We first tested whether docetaxel is an agent that can increase time to recovery from shock-induced seizure-like behaviors in *C. elegans.* Acute exposure to 0.005, 0.01, and 1mM docetaxel significantly increased time to recovery [Fig 1A] and percent non-recovery (%NR) [Fig 1B] following electroshock, as compared to M9 saline.

To simulate seizure-like behavior severity observed in cancer patients undergoing prolonged chemotherapeutic treatments, we chronically exposed the nematodes to varying concentrations of docetaxel. Chronic exposure to docetaxel

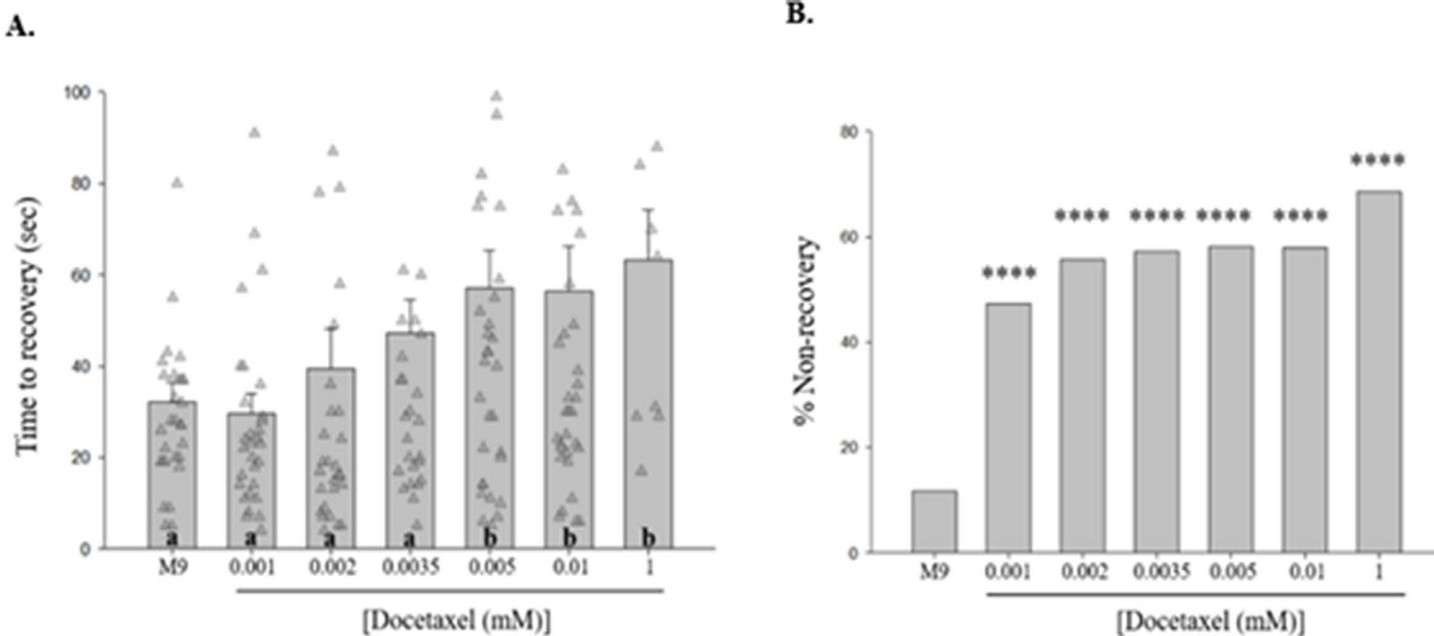

**Fig 1. Acute docetaxel treatment increases time to recovery from shock-induced seizure-like behaviors with increasing concentration in *C. elegans*.** (A) Acute exposure to docetaxel increases time to recovery in a concentration-dependent manner. Different letters denote a statistically significant difference in the mean values between the groups, where "a" stands for not statistically significantly different from M9 saline and "b" stands for statistically significantly different from M9 saline (Student-Newman Keuls, $p < 0.05$). Data shown as mean ± s.e.m. (B) Acute exposure to increasing concentrations of docetaxel increases the percentage of non-recovered worms following the electroshock. 0.001 mM DTX vs. M9, $X^2 = 16.6071$, $p < 0.0001$; 0.002 mM DTX vs. M9, $X^2 = 18.7615$, $p < 0.0001$; 0.0035 mM DTX vs. M9, $X^2 = 19.3063$, $p < 0.0001$; 0.005 mM DTX vs. M9, $X^2 = 21.698$, $p < 0.0001$; 0.01 mM DTX vs. M9, $X^2 = 20.7299$, $p < 0.0001$; 1 mM DTX vs. M9, $X^2 = 24.2858$, $p < 0.0001$. $N > 30$ for each group. ****, $p < 0.0001$, compared to M9.

increased time to recovery from shock-induced seizure-like behaviors [Fig 2A] and % NR [Fig 2B], in contrast to nematodes exposed to M9 saline, which did not exhibit such changes.

Sildenafil citrate (SC), the generic formulation of Viagra®, indirectly activates potassium channel conductance [23]. Acute treatment with 0.5 mM SC had no effect on time to recovery [Fig 3A]. However, when combined with acute exposure to 0.01 mM docetaxel, SC significantly suppressed the time to recovery at concentrations as low as 0.06 mM, although the protective effect was not statistically significant at the two highest concentrations of SC [Fig 3A]. Acute exposure to 0.5 mM SC on its own increased % NR over M9 saline [Fig 3B]. Additionally, co-administration of SC, at concentrations as low as 0.06 mM, suppressed the increased % NR following acute 0.01 mM docetaxel [Fig 3B].

Resveramorph-3 (RVM-3) is a member of a family of bridged bicyclic compounds inspired by resveratrol [22]. Based on its structure and similarity to related compounds, RVM-3 has been proposed to function as an irreversible agonist, although its mechanism of action has not been established [24]. In our experiments, 100 µM RVM-3, when combined with acute exposure to 0.01 and 1 mM docetaxel, significantly reduced shock-induced seizure-like behavior duration [Fig 4A] and % NR [Fig 4B].

When combined with chronic exposure to 0.005 mM docetaxel, 0.1 and 0.25 mM acute SC suppressed time to recovery [Fig 5A], and 0.25 mM SC suppressed %NR below control levels [Fig 5B]. Acute treatment with 100 µM RVM-3 significantly decreased time to recovery when the animals were chronically exposed to 0.005 mM docetaxel [Fig 6A]. However, RVM-3 did not suppress the increased % NR with chronic 0.005 mM docetaxel exposure [Fig 6B]. Additionally, acute treatment with 100 µM RVM-3 significantly decreased the time to recovery [Fig 7A] and decreased % NR [Fig 7B] in animals subjected to a chronic treatment of 3.5 or 10 µM docetaxel.

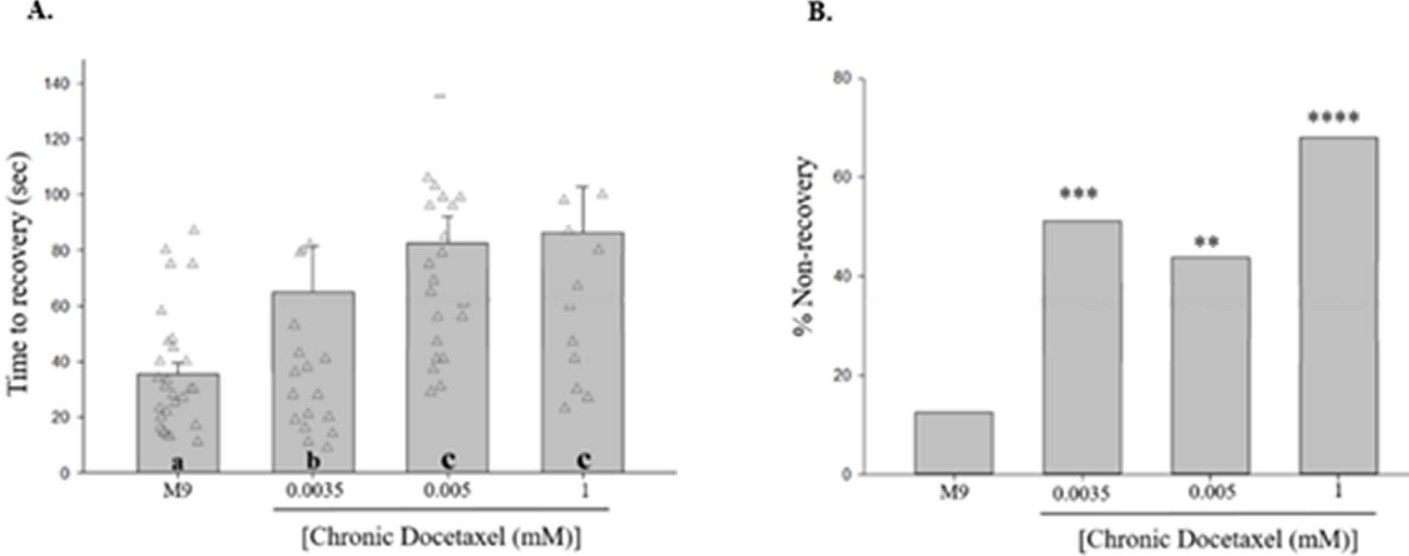

**Fig 2. Nematodes treated with chronic docetaxel, display an increase in time to recovery from shock-induced seizure-like behaviors when compared to nematodes exposed to M9 saline alone. (A)** Chronic exposure to docetaxel increases time to recovery with increasing concentrations. Different letters denote a statistically significant difference in the mean values between the groups where "a" stands for not statistically significantly different from M9 saline, "b" stands for statistically significantly different from M9 saline, and "c" stands for statistically significantly different from M9 saline and solutions labeled "b" (Student-Newman Keuls, $p < 0.05$). Data shown as mean ± s.e.m. **(B)** Chronic exposure to increasing concentrations of docetaxel increases the percentage of non-recovered worms following the electroshock. **C.**E. 0.0035 mM DTX vs. M9, $X^2 = 11.9656$, $p = 0.0005$; **C.**E. 0.005 mM DTX vs. M9, $X^2 = 8.4176$, $p = 0.0037$; **C.**E. 1 mM DTX vs. M9, $X^2 = 24.7284$, $p < 0.0001$. $N > 30$ for each group. The horizontal reference line (0.01 mM Docetaxel Acute Exposure) indicates the mean value observed for acute exposure to 0.01 mM docetaxel [Fig 1]. **, $p < 0.01$; ***, $p < 0.001$; ****, $p < 0.0001$; all compared to M9.

## Discussion

Docetaxel kills cancerous cells by disrupting the dynamic assembly of polymer microtubule subunits, which leads to Docetaxel-Induced Peripheral Neuropathies (DIPNs) [25]. In severe cases, DIPN-induced muscle weakness often leads patients to abandon their chemotherapy regimen, and there is an urgent need for effective agents and/or therapies that mitigate and/or prevent the development of DIPNs. In this study, we acutely or chronically exposed *C. elegans* to docetaxel, demonstrating that the drug can modulate time to recovery from shock-induced seizure-like behaviors in the invertebrate model. Based on our findings we can hypothesize that nematodes treated with acute docetaxel exposure displayed an increase in time to recovery from shock-induced seizure-like behaviors [Fig 1A]. We attribute drug effects seen from acute exposure to be concentration dependent such that after reaching threshold, drug toxicity led to the display of increased time to recovery from shock-induced seizure-like behaviors in nematodes. Regarding nematodes treated with chronic docetaxel exposure, we observed increased time to recovery from shock-induced seizure-like behaviors, unlike nematodes exposed to M9 saline, which did not exhibit such behaviors [Fig 2A]. In this scenario, we attribute drug effects to be due to time dependent toxicity with sensitization, not purely concentration dependent effects.

The NO-cGMP/PKG pathway is a potential target for seizure therapeutics, as it influences downstream K$^+$ channel conductance through key players like PKG and PP2A [23]. Additionally, repurposing of sildenafil citrate has gained significant interest within the pharmaceutical and healthcare communities, as it shows potential to improve drug tolerability and efficacy of chemotherapy [26]. In this study, we show that acute treatment with sildenafil citrate, a PKG pathway activator [23], significantly reduces time to recovery from shock-induced seizure-like behaviors following acute [Fig 3A] and chronic [Fig 5A] exposure to docetaxel.

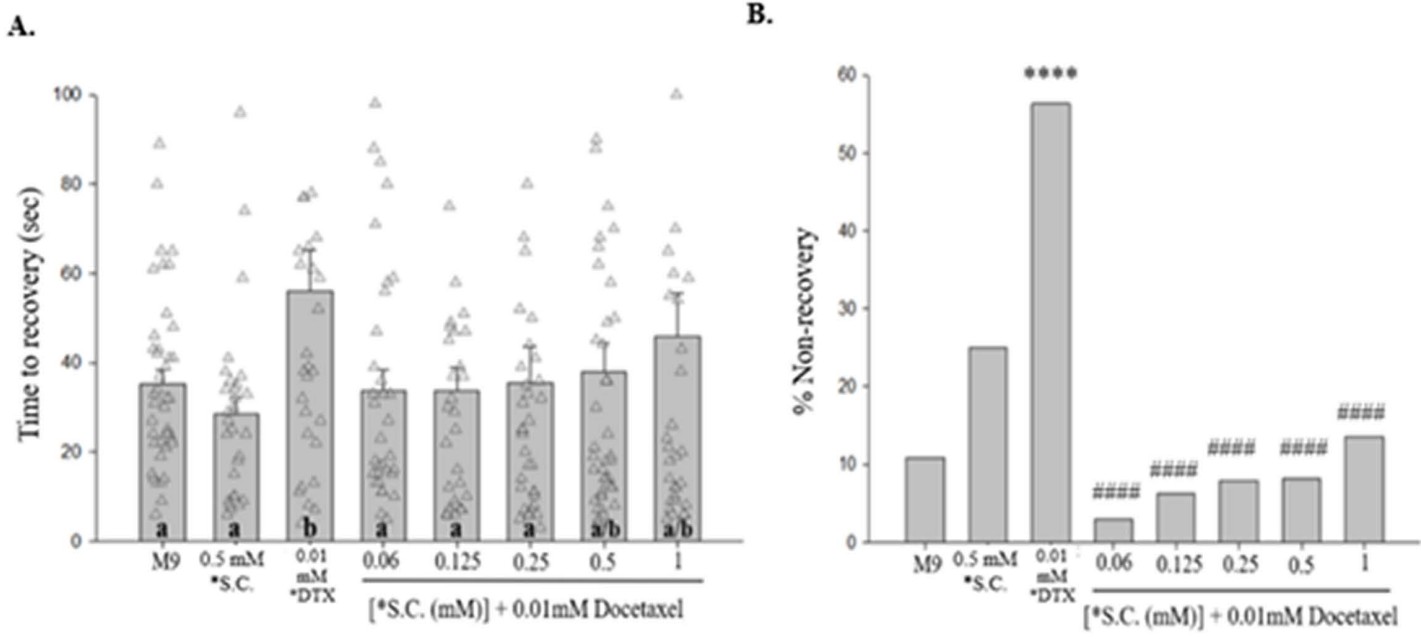

*S.C.: Sildenafil Citrate

*DTX: Docetaxel

**Fig 3. Acute treatment with sildenafil citrate decreases the duration of shock-induced seizure-like behaviors that accompany acute exposure to docetaxel. (A)** Acute treatment with sildenafil citrate significantly decreases time to recovery for worms acutely treated with 0.01 mM docetaxel. Different letters denote a statistically significant difference in the mean values between the groups where "a" stands for not statistically significantly different from M9 saline, "b" stands for statistically significantly different from M9 saline, and "a/b" stands for not statistically significantly different from M9 saline or solutions labeled "b" (Student-Newman Keuls, $p < 0.05$). Data shown as mean ± s.e.m. **(B)** Acute treatment with sildenafil citrate decreases the percentage of non-recovered worms following the electroshock. 0.5 mM **S.C.** vs. M9, $X^2 = 2.9668$, $p = 0.0850$; 0.01 mM DTX vs. M9, $X^2 = 25.0325$, $p < 0.0001$; 0.01 mM DTX. vs. 0.06 mM **S.C.** + 0.01 mM DTX, $X^2 = 28.1521$, $p < 0.0001$; 0.01 mM DTX vs. 0.125 mM **S.C.** + 0.01 mM DTX, $X^2 = 23.4833$, $p < 0.0001$; 0.01 mM DTX vs. 0.25 mM **S.C.** + 0.01 mM DTX, $X^2 = 24.9289$, $p < 0.0001$; 0.01 mM DTX vs. 0.5 mM **S.C.** + 0.01 mM DTX, $X^2 = 29.0098$, $p < 0.0001$; 0.01 mM DTX vs. 1 mM **S.C.** + 0.01 mM DTX, $X^2 = 18.9077$, $p < 0.0001$. $N > 30$ for each group. ****, $p < 0.0001$, compared to M9. ####, $p < 0.0001$, compared to 0.01 mM DTX. **S.C.**, Sildenafil Citrate; DTX, Docetaxel.

Docetaxel inhibits K+ currents in a dose-dependent manner [27], whereas sildenafil citrate activates the NO/cGMP/PKG pathway, possibly promoting K+ conductance as speculated by previous work in our lab [23]. Furthermore, the effects of sildenafil have been linked to the GABAergic system [28], suggesting that sildenafil citrate reduces neuronal excitability through mechanisms involving both K+ channels and GABA signaling, ameliorating DIPN-related effects. Being a PDE5-inhibitor, sildenafil works as a vasodilator, following the increase in cyclic guanosine (cGMP) levels, leading to smooth muscle relaxation and vasodilation [28] which can promote neuroprotective effects. In this study, sildenafil citrate significantly reduced recovery time following shock-induced seizure-like behaviors [Fig 3, 5], consistent with prior findings that sildenafil citrate decreases recovery time in this assay [29]. Based on these observations, we interpret sildenafil citrate as a compound that reduces recovery time in this behavioral paradigm without evidence of altering shock induction. In terms of toxicity, from our findings [Fig 3, 5] we observed that acute co-exposure to high-dose sildenafil citrate and 0.01 mM docetaxel increased seizure-like behaviors, whereas chronic docetaxel exposure followed by acute sildenafil citrate treatment did not produce this effect. This pattern suggests a time-dependent interaction in which simultaneous exposure may transiently heighten sensitivity to seizure-like behaviors, while chronic docetaxel exposure may induce adaptive responses that mitigate this sensitivity. This underlying mechanism remains unclear and warrants further investigation.

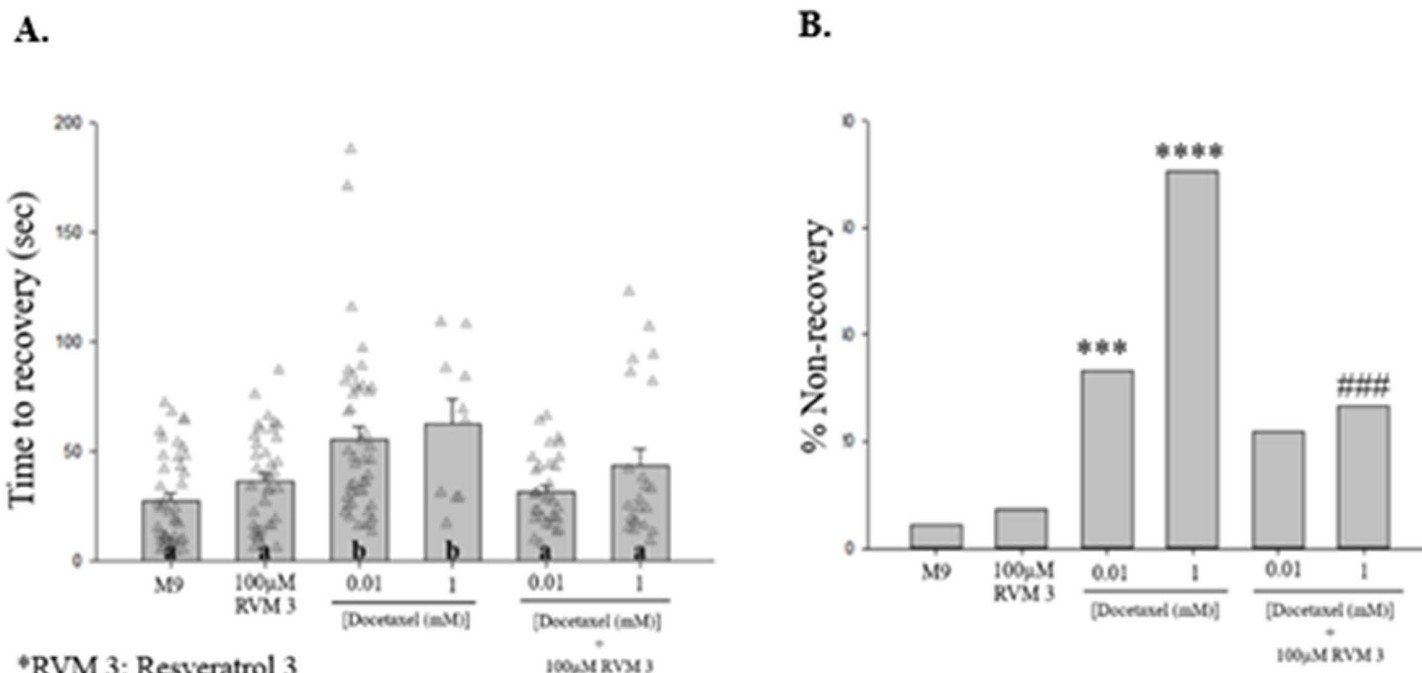

**Fig 4. Acute treatment with RVM-3 decreases the duration of shock-induced seizure-like behaviors that accompany acute exposure to docetaxel. (A)** Acute treatment with RVM-3 significantly decreased time to recovery for worms treated with 0.01 and 1 mM docetaxel. Different letters denote a statistically significant difference in the mean values between the groups where "a" stands for not statistically significantly different from M9 saline, and "b" stands for statistically significantly different from M9 saline (Student-Newman Keuls, $p < 0.05$). Data shown as mean ± s.e.m. **(B)** Acute treatment with 100 μM RVM-3 significantly decreases the percentage of non-recovered worms treated with 1 mM docetaxel. 100 μM RVM3 vs. M9, $X^2 = 0.3233$, $p = 0.5696$; 0.01 mM DTX vs. M9, $X^2 = 13.1767$, $p < 0.0001$; 1 mM DTX vs. M9, $X^2 = 38.3749$, $p < 0.0001$; 0.01 mM DTX vs. 0.01 mM DTX + 100 μM RVM3, $X^2 = 1.5921$, $p = 0.2070$; 1 mM DTX vs. 1 mM DTX + 100 μM RVM3, $X^2 = 12.2982$, $p < 0.0005$. $N > 30$ for each group. ***, $p < 0.001$; ****, $p < 0.0001$; all compared to M9. ###, $p < 0.001$, compared to 1 mM DTX. RVM-3, Resveramorph 3.

Moreover, resveratrol-inspired compounds, known as Resveramorphs (RVM), possess a three-dimensional structure that increases complementarity to a binding site leading to improved target selectivity and suggesting potential neuro-protective effects [22,24]. Our lab has previously demonstrated that in *C. elegans,* shock-induced seizure-like behaviors decreased in duration when worms were exposed to acute combinations of Resveramorph-3 (RVM-3) and pentylenetetra-zol (PTZ) solutions [24]. In our study, acute exposure to 100 μM RVM-3 in combination with docetaxel decreased the duration of shock-induced seizure-like behaviors [Fig 4]. Nematodes grown chronically in docetaxel and then treated acutely with RVM-3 showed a similar reduction [Fig 6]. These findings suggest that RVM-3 can reduce recovery time from seizure-like behaviors under both acute and chronic conditions. One interpretation is that the pathways or processes influenced by RVM-3 remain responsive regardless of prior docetaxel exposure; however, the basis of these effects is not yet understood. RVM-3 is a novel compound and therefore further studies are required to evaluate its potential toxicity, define dose-dependent effects, and determine whether the observed reductions in recovery time reflect specific actions within this behavioral paradigm or broader effects on neuronal or locomotor function.

The mechanism by which docetaxel modulates time to recovery from shock-induced seizure-like behaviors in *C. elegans* is not clearly defined in this study, as electrophysiological changes during drug exposure were not examined. While there is no direct evidence linking docetaxel to axonal or neuronal degeneration in nematodes, prior studies show taxol-induced mecha-notoxicity [17], therefore, we speculate the possibility of similar axonal damage from docetaxel. Docetaxel-induced neuropathy is typically mild to moderate, but high cumulative doses can cause severe clinical and electrophysiological abnormalities, with

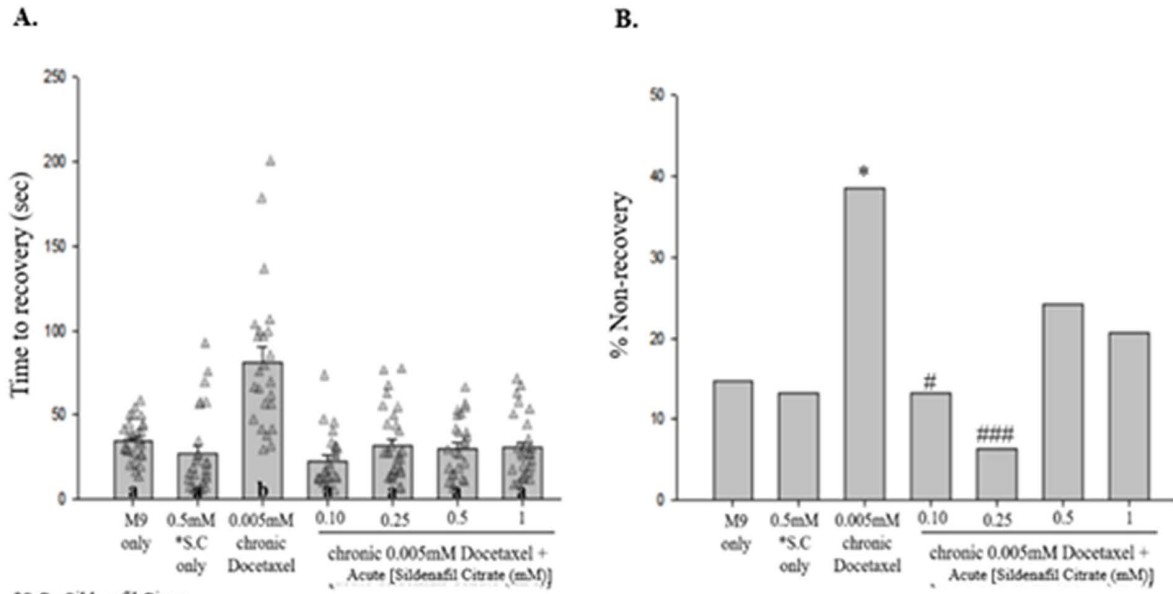

**Fig 5. Acute treatment with sildenafil citrate decreases the duration of shock-induced seizure-like behaviors that accompany chronic exposure to docetaxel. (A)** Acute treatment with various concentrations of sildenafil citrate significantly decreases time to recovery for worms treated with chronic 0.005 mM docetaxel. Different letters denote a statistically significant difference in the mean values between the groups where "a" stands for not statistically significantly different from M9 saline, and "b" stands for statistically significantly different from M9 saline (Student-Newman Keuls, $P < 0.05$). Data shown as mean ± s.e.m. **(B)** Acute treatment with 0.10 and 0.25 mM sildenafil citrate decreases percent non-recovery following electroshock of nematodes exposed to chronic 0.005 mM docetaxel, while higher concentrations (0.5 and 1mM) were less effective. 0.5 mM **S.**C. vs. M9, $X^2 = 0.0248$, $p = 0.8749$; 0.005 mM DTX vs. M9, $X^2 = 5.1534$, $p = 0.0232$; **C.**E. 0.005 mM DT vs. 0.1 mM **S.**C. + **C.**E. 0.005 mM DTX, $X^2 = 5.3657$, $p = 0.0205$; **C.**E. 0.005 mM DTX vs. 0.25 mM **S.**C. + **C.**E. 0.005 mM DTX, $X^2 = 9.6243$, $p = 0.0019$; **C.**E. 0.005 mM DTX vs. 0.5 mM **S.**C. + **C.**E. 0.005 mM DTX, $X^2 = 1.6623$, $p = 0.1973$; **C.**E. 0.005 mM DTX vs. 1 mM **S.**C. + **C.**E. 0.005 mM DTX, $X^2 = 2.756$, $p = 0.0969$. $N > 30$ for each group. *, $p < 0.05$, compared to M9. #, $p < 0.05$; ###, $p < 0.001$; all compared to 0.005 mM **C.**E. DTX. **S.**C., Sildenafil Citrate.

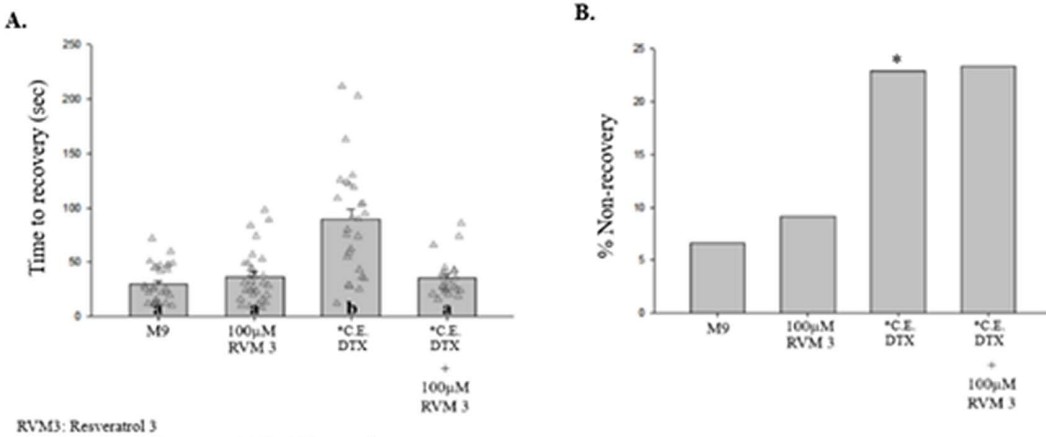

**Fig 6. Acute treatment with RVM 3 decreases the duration of shock-induced seizure-like behaviors that accompany chronic exposure to docetaxel. (A)** Acute treatment with RVM 3 significantly decreases time to recovery for worms treated with chronic 0.005 mM docetaxel. Different letters denote a statistically significant difference in the mean values between the groups where "a" stands for not statistically significantly different from M9 saline, and "b" stands for statistically significantly different from M9 saline (Student-Newman Keuls, $P < 0.05$). Data shown as mean ± s.e.m. **(B)** Acute treatment with RVM 3 does not decrease the percentage of non-recovered worms subjected to chronic 0.005 mM docetaxel exposure. 100 μM RVM3 vs. M9, $X^2 = 0.1079$, $p = 0.7425$; **C.**E. 0.005 mM DTX vs. M9, $X^2 = 3.6308$, $p = 0.0567$; **C.**E. 0.005mM DTX vs. 100 μM RVM3 + **C.**E. 0.005 mM DTX, $X^2 = 0.0021$, $p = 0.9634$. *, $p < 0.05$, compared to M9. RVM-3, Resveramorph 3; **C.**E DTX, chronic exposure to 0.005 mM docetaxel.

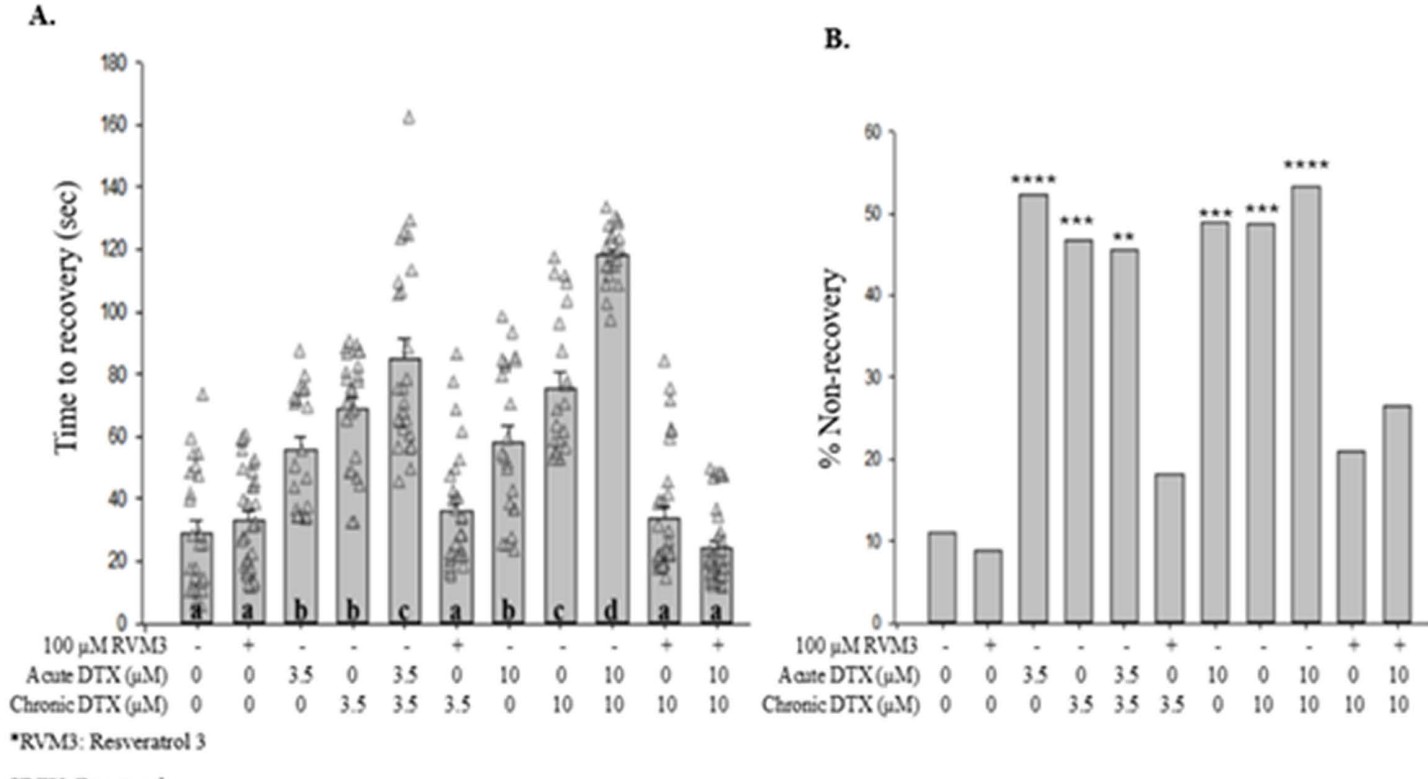

**Fig 7. Acute treatment with RVM-3 decreases the duration of shock-induced seizure-like behaviors associated with acute and chronic exposure to docetaxel. (A)** Acute treatment with RVM-3 significantly decreased time to recovery in worms treated with acute and/or chronic 0.0035 and 0.01 mM docetaxel. Different letters denote a statistically significant difference in the mean values between the groups where "a" stands for not statistically significantly different from M9 saline, "b" stands for statistically significantly different from M9 saline, "c" stands for statistically significantly different from M9 saline and other labeled solutions, and "d" stands for statistically significantly different form M9 saline and other labeled solutions (Student-Newman Keuls, $P < 0.05$). Data shown as mean ± s.e.m. **(B)** Acute treatment with RVM-3 decreased %NR in animals treated with chronic 0.0035 and 0.01 mM docetaxel, following the electroshock. 100 µM RVM3 vs. M9, $X^2 = 0.0888$, $p = 0.7657$; Acute 0.0035 mM DTX vs. M9, $X^2 = 12.1153$, $p < 0.0001$; **C.E.** 0.0035 mM DTX in M9 vs. M9, $X^2 = 9.7686$, $p = 0.0018$; **C.E.** 0.0035 mM DTX in 0.0035 mM DTX vs. M9, $X^2 = 9.1983$, $p = 0.0024$; **C.E.** 0.0035 mM DTX in 100 µM RVM3 vs. M9, $X^2 = 0.5822$, $p = 0.4455$; Acute 0.01 mM DTX vs. M9, $X^2 = 10.4772$, $p = 0.0012$; **C.E.** 0.01 mM DTX in M9 vs. M9, $X^2 = 10.3204$, $p = 0.0013$; **C.E.** 0.01 mM DTX in 0.01 mM DTX vs. M9, $X^2 = 12.8355$, $p < 0.0001$; **C.E.** 0.01 mM DTX in 100 µM RVM3 vs. M9, $X^2 = 1.2115$, $p = 0.2710$; **C.E.** 0.01 mM DTX in 0.01 mM DTX + 100 µM RVM3 vs. M9, $X^2 = 1.1258$, $p = 0.2887$. $N > 30$ for each group. **, $p < 0.01$; ***, $p < 0.001$; ****, $p < 0.0001$; all compared to M9. RVM-3, Resveramorph 3; DTX, Docetaxel.

nerve biopsies in humans showing large, myelinated fiber loss and occasional axonal degeneration [3]. Future studies could test the effects of PKG activators on other proconvulsant such as PTZ or glyphosate-based compounds [18].

Future work may validate the involvement of the NO/cGMP/PKG pathway by using a secondary PDE5 inhibitor, such as tadalafil, or using genetic mutants of the PKG homolog *egl-4* [29], or of relevant cGMP phosphodiesterases (*pde-1, pde-5*) [23]. Additionally, replicating the anticonvulsant properties of sildenafil citrate and RVM-3 in mammalian models of seizure-like behaviors will prove useful for assessing their potential as a combinatorial therapeutic. Inclusion of full dose-response model fitting curves and associated EC50/IC50 values would provide additional quantitative insight. However, the primary aim of this study is to establish and demonstrate the utility of our electroshock assay as a screening platform rather than to generate complete pharmacological profiles for each compound. In this study, the concentration-dependent assessments are qualitative and exploratory, therefore, we acknowledge that there is a need for future studies to perform a full-dose response characterization.

## Supporting information

**S1 Fig. One-way ANOVA: Acute docetaxel treatment increases time to recovery from shock-induced seizure-like behaviors with increasing concentration in *C. elegans.***
(TIF)

**S2 Fig. Chi-square test: Acute docetaxel treatment increases time to recovery from shock-induced seizure-like behaviors with increasing concentration in *C. elegans.***
(TIF)

**S3 Fig. One-way ANOVA: Nematodes treated with chronic docetaxel, display an increase in time to recovery from shock-induced seizure-like behaviors when compared to nematodes exposed to M9 saline alone.**
(TIF)

**S4 Fig. Chi-square test: Nematodes treated with chronic docetaxel, display an increase in time to recovery from shock-induced seizure-like behaviors when compared to nematodes exposed to M9 saline alone.**
(TIF)

**S5 Fig. One-way ANOVA: Acute treatment with sildenafil citrate decreases the duration of shock-induced seizure-like behaviors that accompany acute exposure to docetaxel.**
(TIF)

**S6 Fig. Chi-square test: Acute treatment with sildenafil citrate decreases the duration of shock-induced seizure-like behaviors that accompany acute exposure to docetaxel.**
(TIF)

**S7 Fig. One-way ANOVA: Acute treatment with RVM-3 decreases the duration of shock-induced seizure-like behaviors that accompany acute exposure to docetaxel.**
(TIF)

**S8 Fig. Chi-square test: Acute treatment with RVM-3 decreases the duration of shock-induced seizure-like behaviors that accompany acute exposure to docetaxel.**
(TIF)

**S9 Fig. One-way ANOVA: Acute treatment with sildenafil citrate decreases the duration of shock-induced seizure-like behaviors that accompany chronic exposure to docetaxel.**
(TIF)

**S10 Fig. Chi-square test: Acute treatment with sildenafil citrate decreases the duration of shock-induced seizure-like behaviors that accompany chronic exposure to docetaxel.**
(TIF)

**S11 Fig. One-way ANOVA: Acute treatment with RVM-3 decreases the duration of shock-induced seizure-like behaviors that accompany chronic exposure to docetaxel.**
(TIF)

**S12 Fig. Chi-square test: Acute treatment with RVM-3 decreases the duration of shock-induced seizure-like behaviors that accompany chronic exposure to docetaxel.**
(TIF)

**S13 Fig. One-way ANOVA: Acute treatment with RVM-3 decreases the duration of shock-induced seizure-like behaviors associated with acute and chronic exposure to docetaxel.**
(TIF)

**S14 Fig. Chi-square test: Acute treatment with RVM-3 decreases the duration of shock-induced seizure-like behaviors associated with acute and chronic exposure to docetaxel.**
(TIF)

**S15 Fig. DMSO effects on *C. elegans* using the electroshock assay. A DMSO sham concentration curve at 0.1% and 0.5%, demonstrates these concentrations have no significant effect on wild-type recovery time after electric shock.**
(TIF)

**S16 Fig. RVM-3 concentration recovery comparisons. Recovery time for N2 worms in M9 saline or in the presence of PTZ.**
(TIF)

## Acknowledgments

We thank members of the Dawson-Scully Lab for feedback on the writing of the manuscript. *C. elegans* strains were provided by the CGC, which is funded by NIH Office of Research Infrastructure Programs (P40 OD010440).

## Author contributions

**Conceptualization:** Scarlet J Park.

**Formal analysis:** Crystal Lloyd.

**Investigation:** Paola Ximena Gonzalez-Lerma.

**Methodology:** Paola Ximena Gonzalez-Lerma, Ken Dawson-Scully.

**Project administration:** Paola Ximena Gonzalez-Lerma, Ken Dawson-Scully.

**Resources:** Ken Dawson-Scully.

**Supervision:** Ken Dawson-Scully.

**Validation:** Ken Dawson-Scully.

**Visualization:** Crystal Lloyd, Scarlet J Park.

**Writing – original draft:** Paola Ximena Gonzalez-Lerma.

**Writing – review & editing:** Paola Ximena Gonzalez-Lerma, Scarlet J Park.

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
