## [Decision Letter · Decision Letter 0]

6 Nov 2025

Dear Dr. Gonzalez-Lerma,

We look forward to receiving your revised manuscript.

Kind regards,

Michael Massiah

Academic Editor

PLOS ONE

Journal Requirements:

Additional Editor Comments:

Thank you for submitting your manuscript to PLOS ONE. I also apologize for the tardiness in my response. After careful consideration, we feel that it has merit but does not fully meet PLOS ONE’s publication criteria as it currently stands. Therefore, we invite you to submit a revised version of the manuscript that addresses the points raised during the review process. As you can see from the reviewers comments, there are some concerns that hopefully you can address.

Reviewer's Responses to Questions

**Comments to the Author**

1. Is the manuscript technically sound, and do the data support the conclusions?

Reviewer #1: No

Reviewer #2: Partly

Reviewer #3: Yes

2. Has the statistical analysis been performed appropriately and rigorously?

Reviewer #1: No

Reviewer #2: Yes

Reviewer #3: Yes

3. Have the authors made all data underlying the findings in their manuscript fully available?

Reviewer #1: Yes

Reviewer #2: Yes

Reviewer #3: Yes

4. Is the manuscript presented in an intelligible fashion and written in standard English?

Reviewer #1: Yes

Reviewer #2: Yes

Reviewer #3: Yes

Reviewer #1: Overview

This manuscript investigates the effects of a chemotherapeutic, docetaxel, in an electroconvulsive assay using the model organism C. elegans. Additionally, the authors identify two drugs that ameliorate the docetaxel effects – sildenafil citrate and resveramorph-3.

Overall, the results appear convincing, although there are some concerns about experimental protocols and phrasing that need correcting.

Comments

Acute exposure. What is the final concentration of DMSO for the docetaxel and sildenafil citrate that the worm experiences? It is not clear from the methods whether an equivalent volume of DMSO is used in the controls. DMSO itself can have neuroprotective properties at low concentrations and needs to be included in your controls.

Chronic exposure. With the chronic exposure experiments, however, you are dissolving solutions into your NGM plate. Therefore the chronic drug concentration experienced by worm will by reduced by a factor dependent on the volume of the plate. If your plate has 10 mls of NGM in it, the stated concentrations of docetaxel will be reduced by a factor of 10. And that assumes that the plate volume is standardised (is it?) and that the drug efficacy is not time-dependent (is it?). The authors need to clarify their experimental protocol and what the actual stated concentrations for the chronic experiments were.

Additionally, the wording in Figure 2 implies that chronic treatment “increases severity of behaviours”. But the statistics appear to be done only against chronic treatment of M9. Comparing Figures 1 and 2 – the data look pretty similar, indicating a lack of chronic response. Although given the point above about the chronic concentrations, this experiment may simply be replicating the acute effects. Do the authors have any evidence for an increase in behavioural severity?

Phrasing. The authors do not measure seizures or induction of seizures – they are specifically measuring time-to-recovery or lack of recovery – which is different. In a number of places the authors have misconstrued the experimental outcomes, which needs to be corrected. Specifically:

• Line 140. A proconvulsant is an agent that initiates or lowers the threshold for seizures. The authors should either test for a proconvulsant effect (eg. Identify whether the voltage threshold for seizures is actually lowered) or reword for what is actually shown.

• Line 176 states that “docetaxel models DIPN-seizure behaviour in the worm”. My understanding is that docetaxel is a proconvulsant and there isn’t evidence that it affects time-to-recovery from seizures. This statement needs to be supported or reworded.

• Line 197 specifically states “docetaxel induces seizure-like behaviors in C. elegans”, which is not supported by the results and needs to be reworded.

Minor comments

• Methods line 94. The day after L4-stage worms would be Day 0 Adulthood (not Day 1).

• The rationale for some experiments are not well explained. It isn’t clear to this reader why a neuroprotective effect of Viagra was expected. The authors should justify their choice of putative neuroprotective compounds.

• Figure legends state that “different letters denote a statistically significant difference in the mean values between the Groups” – but then it isn’t explained what “a” and “b” (and later on “a/b”) actually represent. I assume “a” is not significant and “b” is?

• Figure 2. It is unclear why a line for 0.01 mM docetaxel is included when that isn’t being tested chronically.

• Figure 7. It isn’t completely clear here why 3.5 uM is being studied. Nor why it is listed as uM rather than mM (everywhere else in the manuscript).

Reviewer #2: The authors use the C. elegans electroshock paradigm to model docetaxel-associated neurotoxicity and test two anticonvulsant candidates: sildenafil citrate (a PDE-5 inhibitor) and a novel bridged bicyclic compound, Resveramorph-3 (RVM-3). Acute and chronic docetaxel exposure prolong recovery from shock-induced convulsions and increase non-recovery (NR). Sildenafil and RVM-3 reduce seizure-like duration under several conditions. The manuscript is clearly written and addresses a clinically relevant problem (chemotherapy-related neurotoxicity) with a tractable invertebrate model.

Major issues

1. The electroshock assay treats ~6 worms together per tube/shock. Report the number of independent tubes and independent experimental days per condition.

2. Acute test solutions contain 0.1% DMSO (drug dissolved in 1% DMSO then 1:10 in M9), but the stated acute control is M9 saline alone. Please include (and analyze against) an acute 0.1% DMSO vehicle control to match drug conditions; otherwise DMSO effects cannot be excluded.

3. Docetaxel is hydrophobic; “coating NGM plates with M9 containing docetaxel” raises concerns about uniformity and stability. Please specify: volumes applied per plate, drying time, storage/light protection, time from coating to use, and evidence the compound distributes/retains bioactivity on agar. Similarly, RVM-3 is said to be dissolved directly in M9—provide rationale/solubility data, stock concentration, purity, and quality control. These details are critical for reproducibility.

4. The recovery criterion (three sinusoidal waves) is clear, but please state whether scorers were blinded to condition and how worms/tubes were randomized across treatments and days.

5. You repeatedly link seizures to Docetaxel-Induced Peripheral Neuropathies (DIPNs). Seizures are central phenomena; while docetaxel-associated encephalopathy and seizures are reported, seizures are not typically considered a feature of peripheral neuropathy per se. Please reframe as docetaxel-associated neurotoxicity (which can include seizures) rather than “DIPN-related seizures,” unless you provide primary evidence/mechanism tying the peripheral pathology to seizure generation.

6. The Results text states that RVM-3 significantly reduced %NR with both 0.01 mM and 1 mM acute docetaxel, but the figure legend/statistics indicate significance only at 1 mM (0.01 mM + RVM-3 p≈0.21, n.s.).

7. You describe concentration dependence qualitatively. Please include dose–response curves with fitted models where possible (even exploratory), report EC50/IC50 or at least slope parameters, and show per-tube datapoints.

Minor issues

• Mechanistic claims: Tone down language implying mechanism (e.g., “RVM-3 potentially acts as an irreversible agonist by covalently binding…”) unless direct evidence is presented here. Likewise, sildenafil’s effects on K⁺ channels are likely indirect via cGMP/PKG; keep causal language cautious and clearly attributed to prior work.

• Terminology consistency: Use “seizure-like behavior” throughout for C. elegans; avoid unqualified “seizures”. Ensure DTX not “DTC” (typo appears once in a legend) and standardize µM/mM spacing.

• Citations: Consider citing C. elegans cGMP/PKG genetics (e.g., egl-4; endogenous pde genes), to support the proposed pathway testing in future work.

Reviewer #3: The use of C. elegans to screen docetaxel effects is interesting and aids in addressing the effects of this chemotherapeutic agent and means of reducing the effects of seizures and altering the recovery time from a seizure by Resveramorph-3 and sildenafil citrate. One view by researchers, in general, is to show the effects of compounds on as many animal models as possible to examine various potential therapeutic medications as one can learn more about the mechanisms of actions including the potential side effects. This study is done well with very detailed explanations in the methodology. The supplemental data is detailed along with the statistical analysis performed.

There are just a few suggestions for the authors which I fell will help a reader better understand overview of the study and why particular compounds are used.

Minor suggestions:

1. Without having to read papers cited( 18-21 ) can the authors give a little bit of background in this manuscript why electroshock is a good model to induce seizures.

2. I am not sure of the logic to try Sildenafil citrate as compared to more specific seizure reducing medications like levetiracetam or the various other medications. The authors state Sildenafil citrate works in the C. elegans model to reduce the seizure induced by docetaxel. But does Sildenafil citrate reduce the induction of the electroshock itself ? So then the seizure is already dampened to reduce the effect of the electroshock but not necessarily the effect of docetaxel ?

3. In the Discussion it is stated that Docetaxel-Induced Peripheral Neuropathies (DIPNs) while will also cause seizures- related to humans.

So, it is not known if Docetaxel induces seizures in C. elegans or did I miss something ? Or are the seizures only induced by electroshock in this model with ones exposed to Docetaxel?

4. It could strengthen the article by including some discussion about the potential side effects or toxicity of sildenafil citrate and Resveramorph-3 in C. elegans, since understanding any harmful or side effects is important when considering these compounds as treatments.

**Do you want your identity to be public for this peer review?** For information about this choice, including consent withdrawal, please see our Privacy Policy

Reviewer #1: No

Reviewer #2: No

Reviewer #3: No

---

## [Author Response · Author response to Decision Letter 1]

16 Dec 2025

Dear Reviewers,

We want to thank the reviewers for their thoughtful and constructive feedback. We have carefully considered each comment and revised the manuscript accordingly. We believe that the added clarifications and details have strengthened the manuscript. Below, we provide our detailed responses to each point.

Reviewer 1:

Major Comments:

Acute exposure

1. What is the final concentration of DMSO for the docetaxel and sildenafil citrate that the worm experiences? It is not clear from the methods whether an equivalent volume of DMSO is used in the controls.

In the acute condition exposures, we diluted docetaxel and sildenafil citrate in 1% dimethyl sulfoxide (DMSO) and subsequently diluted 1:10 in M9 saline. This methodology was also used for chronic condition exposures. This means that for acute condition exposures, the tested solutions contained M9 with 0.1% DMSO, meaning 0.1% of the total volume is DMSO and the remaining 99.9% is M9 saline. We note this detail under “chronic exposure protocol” (Lines 107-111). We also briefly reference this under the section titled “drugs and chemicals used” (Lines 131-136), however we acknowledge we did not clarify this for the acute exposure protocol. We appreciate your attention to detail, as we have included this detailed description under the “acute exposure protocol” section (Lines 102-105).

2. DMSO itself can have neuroprotective properties at low concentrations and needs to be included in your controls.

Thank you for pointing this out. When we implemented DMSO as a dissolving vehicle for our compounds, we also considered the possibility that low concentrations of DMSO could generate neuroprotective effects, therefore, to address this issue, we turned to previously published data by our lab. In previous work, a concentration curve for dimethyl sulfoxide (DMSO) was conducted showing that there is no significant difference in recovery from electric shock in concentrations up to 0.5% DMSO [19]. We have included this dose curve image as a supplemental figure titled S8 Fig. DMSO effects on C. elegans using the electroshock assay. This is important since we accept that DMSO can be used as a solvent for this assay, without affecting the results observed by docetaxel and sildenafil citrate. For further clarification, we make a note of these results in the section titled “drugs and chemicals used” (Lines 131-136). We have also included the figure in this response for your accessibility. Considering that we had previously published work showing that a 0.1% concentration of DMSO did not show significant effects on the recovery-time of wild-type nematodes after electric shock, we did not run extra sets of controls in our work. If you feel like this is something that must be included in our current work, we would be happy to run the extra set of experiments. We like to note that we respectfully value your feedback on this matter and we seek to only justify our original methodology for this part of our study. However, if you do feel this is something that must be repeated we will address it without a problem.

Note: A DMSO sham concentration curve at 0.1% and 0.5%, and demonstrates these concentrations have no significant effect on wild-type recovery time after electric shock [19].

Chronic exposure

3. With the chronic exposure experiments, however, you are dissolving solutions into your NGM plate. Therefore, the chronic drug concentration experienced by worm will by reduced by a factor dependent on the volume of the plate. If your plate has 10 mls of NGM in it, the stated concentrations of docetaxel will be reduced by a factor of 10. And that assumes that the plate volume is standardised (is it?)

For chronic exposure experiments, we coated each plate with M9 saline containing the corresponding concentrations of docetaxel or vehicle control (M9 with 0.1% v/v dimethyl sulfoxide (DMSO). We mention this in the “chronic exposure protocol” section (Lines 107-109). In more detail, 10 milliliters of NGM agar were first dispensed per plate and once the agar solidified, 2 milliliters of corresponding solutions were poured on the surface of the agar plates. Plates were then covered and left to dry for 24 hours, away from light at room temperature, to avoid light and temperature degradation. All plates used in the chronic exposure protocol were standardized. After plates air-dried with the coated layer of corresponding solution, these were ready to use for experiments. New batches of plates were prepared every week to avoid drug degradation through time. For experiments, worms were placed on the surface of the plates. It is important to note that C. elegans do not burrow into the agar, but rather swim on the surface of the agar. A total of 6 gravid adult nematodes were transferred onto the surface of each coated plate and incubated for 2 days at 20 ℃. After 3 days of incubation, L4-stage worms were then transferred to a new drug-coated plate for overnight incubation at 22 ℃. On day 4, 1-day-old adult worms were incubated for 30 minutes in M9 saline (control) or the test solution prior to the electroshock delivery. We have included this detailed description of the protocol under the “chronic exposure protocol” section (Lines 111-125). Regarding acute docetaxel exposure, we observed that drug effects were concentration-dependent such that after reaching a threshold, the drug generated toxicity to the nematode which we observed as an increase in time-to-recovery from the shock-induced seizure-like behaviors (Lines 323-326). In the scenario of chronic drug exposure, we hypothesize that the effects observed were time-dependent with sensitization, not purely concentration effects. This due to the fact that chronic growth could have pre-disposed the nematode to be then affected by the subsequent acute drug exposure, resulting in toxicity seen as an increase in time-to-recovery from shock-induced seizure-like behaviors starting at lower concentrations (Lines 326-329).

4. The authors need to clarify their experimental protocol and what the actual stated concentrations for the chronic experiments were

We have included a detailed description of the chronic exposure protocol in the above response for Question 3. However, to further clarify, we have poured 2 milliliters of the prepared solutions onto the surface of each solidified agar plate. Plates were then covered and left to air-dry for 24 hours, away from light at room temperature to avoid light and temperature degradation. All plates used in the chronic exposure protocol were standardized. After plates air-dried with the coated layer of corresponding solution, these were ready to use for experiments. Additionally, for drug concentration specification, docetaxel was first dissolved in 1% DMSO, followed by a 1:10 dilution in M9 saline. Therefore, the tested solutions were dissolved in 0.1% DMSO of the total volume with a 99.9% M9 saline. This information was added under the “chronic exposure protocol” section (Lines 107-111).

5. Wording in Figure 2 implies that chronic treatment “increases severity of behaviours”. But the statistics appear to be done only against chronic treatment of M9. Comparing Figures 1 and 2 – the data look pretty similar, indicating a lack of chronic response. Although given the point above about the chronic concentrations, this experiment may simply be replicating the acute effects. Do the authors have any evidence for an increase in behavioural severity?

Figure 2 titled “Chronic docetaxel treatment increases severity of seizure-like behaviors”, is meant to show that nematodes exposed to a chronic docetaxel protocol, tend to display a concentration-dependent increase in time-to-recovery from a seizure-like behavior at lower concentrations (Figure 2), than those nematodes exposed to an acute docetaxel protocol (Figure 1). In figure 1, nematodes exposed to 0.0035 mM acute docetaxel experienced a time-to-recovery from seizure-like behaviors similar to worms exposed to M9 saline. However, nematodes exposed to chronic 0.0035 mM docetaxel experienced an increase in time-to-recovery from seizure-like behaviors that was significantly different from M9 saline. In our figures, we display statistical differences by using different letters. In figure 1, M9 saline and acute 0.0035 mM docetaxel were labeled “a”, indicating that both solutions do not display any significantly difference in time-to-recovery from seizure-like behaviors. In figure 2, M9 saline was labeled “a”, whereas chronic 0.0035 mM docetaxel was labeled “b” indicating a statistically significant difference in time-to-recovery from seizure-like behaviors between the two solutions.

To ensure clarity of the data displayed, we are changing the title for figure 2 to “Nematodes treated with chronic docetaxel, displayed an increase in time-to-recovery from seizure-like behaviors starting at lower concentrations” (see Fig 2 in figure files and manuscript Lines 204-205). In figure 2, we also compared the chronic exposure data to the acute exposure data. In this figure, we show that nematodes in M9 saline, column labeled “a”, displayed a statistically significant decreased time-to-recovery from seizure-like behaviors versus worms treated with chronic 0.0035 mM docetaxel labeled “b”. Furthermore, nematodes treated with chronic 0.005 mM and 1 mM docetaxel displayed a statistically significant increase in time-to-recovery different than animals exposed to chronic 0.0035 mM docetaxel and M9, therefore these columns were labeled “c”. As a form of comparison, we included a horizontal solid line in figure 2 to show that on average, nematodes exposed to acute 0.01 mM docetaxel displayed lower time-to-recovery from seizure-like behaviors than those animals exposed to chronic docetaxel concentrations. We indicate that different letters denote statistically significant difference in the mean values between the groups, and we also explain the meaning of each letter in each figure legend (Lines 190-192, 207-210, 229-231, 251-252, 272-274, 288-289, 299-302).

6. Phrasing. The authors do not measure seizures or induction of seizures – they are specifically measuring time-to-recovery or lack of recovery – which is different. In a number of places the authors have misconstrued the experimental outcomes, which needs to be corrected. Specifically:

We appreciate the reviewer for pointing this out. We agree that by mentioning the increase or decrease of seizure-like behaviors, we are conveying the incorrect idea. In this study, we are measuring time-to-recovery from a shock-induced seizure-like behavior, using our previously established electroshock assay [18-21]. Here, shock induces the seizure-like behavior, while exposure to M9 saline versus acute or chronic docetaxel affects the duration of time-to-recovery from the induced seizure-like behavior. Additionally, we are using sildenafil citrate and resveramorph-3 (RVM-3) to measure the effects these have in decreasing the duration of time-to-recovery from the shock-induced seizure-like behaviors once altered by exposure to docetaxel (Lines 39-42). To clarify our methodology and results, we have addressed each of the scenarios pointed out by the reviewer below:

• Line 140. A proconvulsant is an agent that initiates or lowers the threshold for seizures. The authors should either test for a proconvulsant effect (eg. Identify whether the voltage threshold for seizures is actually lowered) or reword for what is actually shown.

In our “results section, our opening sentence states “We first tested whether docetaxel is a proconvulsant in C. elegans.” As the reviewer mentioned above, a proconvulsant is a substance that promotes convulsions by lowering seizure threshold in the brain, therefore increasing the risk for seizures. In this study, we did not use docetaxel as a proconvulsant, rather we employed the drug to measure its effects in modulating time-to-recovery from shock-induced seizure-like behaviors. We treated our nematodes in both acute and chronic conditions versus M9 saline to observe for the effects that docetaxel had on the time-to-recovery from the shock-induced seizure-like behaviors. To clarify our methodology and findings, we have reworded the phrasing to state “We first tested whether docetaxel is an agent that can increase time-to-recovery from shock-induced seizure-like behaviors in C. elegans” (Lines 183-184).

• Line 176 states that “docetaxel models DIPN-seizure behaviour in the worm”. My understanding is that docetaxel is a proconvulsant and there isn’t evidence that it affects time-to-recovery from seizures. This statement needs to be supported or reworded.

Docetaxel is not a proconvulsant but rather a taxane chemotherapeutic agent used to treat advanced, metastatic, or chemotherapy-resistant cancers, however its neurotoxic side effects often lead patients to abandon treatment [5, 6]. We mention this classification for docetaxel in our manuscript (Lines 34, 55, & 58-61). Docetaxel causes docetaxel-induced peripheral neuropathies (DIPNs) which includes motor neuropathy, tingling, muscle weakness, and numbness rather than acute or chronic seizures directly. This re-wording has been made in our manuscript Lines 35-36. Furthermore, in the past research has concluded that seizures following docetaxel use are rare and are not recognized as acute or chronic toxicity of the drug directly. However, a case study published in 1999 did report a patient developing an encephalopathy and subsequent seizures after receiving docetaxel [9]. This leads us to hypothesize that docetaxel may be used as an agent that may sensitize patients to develop seizures. In our study, we used the electroshock assay to model shock-induced DIPN-related muscle weakness C. elegans. We then treated nematodes with acute and chronic conditions of docetaxel to assess the effects of the drug in modulating time to recovery from shock-induced seizure-like behaviors. We have included this clarification in lines 37-39. Results obtained show that acutely or chronically exposing nematodes to docetaxel increases time to recovery from shock-induced seizure-like behaviors (Lines 37-39). Specifically, nematodes treated with docetaxel exposure displayed an increase in time to recovery from shock-induced seizure-like behaviors [Fig1A]. We attribute these drug effects to be concentration dependent, such that after reaching threshold drug toxicity led to the display of increase in behavior duration (Lines 284-289). For chronic docetaxel exposure, we observed increase time to recovery from shcok-induced seizure-like behaviors starting at lower concentrations [Fig2A]. In this case, we attribute drug effects to be time dependent with sensitization, not purely concentration dependent effects (Lines 328-329).

• Line 197 specifically states “docetaxel induces seizure-like behaviors in C. elegans”, which is not supported by the results and needs to be reworded.

We thank the reviewer for pointing this out. Docetaxel does not trigger seizure-like behaviors in our model. Instead it is the shock the one responsible for inducing the seizure-like behavior, while the docetaxel modulates the duration or time to recovery of the seizure-like behavior. We have re-worded this statement throughout the manuscript and we have also re-worded the sentence above (Lines 201-203).

Minor comments:

7. Methods line 94. The day after L4-stage worms would be Day 0 Adulthood (not Day 1).

We revised our used protocol and we found that there was a typo on the days. On day 1, adult worms with eggs were plated on NGM agar plates seeded with OP50 E. coli for 2 days (rather than 3 days) at 20 ℃. On day 3, L4-stage worms were picked and plated on a new NGM agar plate seeded with OP50 E. coli for overnight incubation at 22 ℃. On day 4, 1-day-old adult worms were incubated for 30 minutes in M9 saline (control) or in the respective test solution prior to the electroshock delivery (Lines 99). To stay consistent with our published protocol for the electroshock assay, we stay consistent with the numbering [20].

8. The rationale for some experiments are not well explained. It isn’t clear to this reader why a neuroprotective effect of Viagra was expected. The authors should justify their choice of putative neuro

---

## [Decision Letter · Decision Letter 1]

16 Jan 2026

Dear Dr. Gonzalez-Lerma,

Thank you for submitting your manuscript to PLOS ONE. As you can see, one of the reviewers still have significant concerns with this manuscript. I hope it is something you can address in a timely manner.  Therefore, we invite you to submit a revised version of the manuscript that addresses the points raised during the review process.

We look forward to receiving your revised manuscript.

Kind regards,

Michael Massiah

Academic Editor

PLOS One

Journal Requirements:

Reviewers' comments:

Reviewer's Responses to Questions

**Comments to the Author**

Reviewer #1: (No Response)

Reviewer #2: All comments have been addressed

2. Is the manuscript technically sound, and do the data support the conclusions?

Reviewer #1: No

Reviewer #2: Yes

3. Has the statistical analysis been performed appropriately and rigorously?

Reviewer #1: No

Reviewer #2: Yes

4. Have the authors made all data underlying the findings in their manuscript fully available?

Reviewer #1: Yes

Reviewer #2: Yes

5. Is the manuscript presented in an intelligible fashion and written in standard English?

Reviewer #1: Yes

Reviewer #2: Yes

Reviewer #1: The authors have addressed most of my comments; however, I still have a couple of major concerns that should be resolved before publication:

Major Comments

1. That’s fine.

2. Essentially that is fine – and I am certainly not going to require you to redo your controls in light of the provided data. But I would suggest in future always doing your controls in the equivalent volume of solvent. I note in the figure provided that the mean is increasing in 0.1% DMSO (from ~35 to 45 sec) with fairly large error bars. And it may well be that the solvent is a contributing factor to the statistical effect seen in this manuscript.

3-4. I am still unclear on the final concentrations for the chronic exposure experiments and think this needs to be clarified explicitly in the text. From the revised wording, the drugs are dissolved in 1% DMSO and then diluted 1/10 in M9 (equivalent to the acute experiments). For the chronic experiments, this is further dissolved in 10 mls NGM (+ 2 mls solution = 12 mls total). The wording certainly implies that the final DMSO concentrations in the chronic experiments will not be 0.1% - they will be ~0.01%. And it should also be clarified that the final drug concentrations are taking the additional volumes into account as well.

5. This is worded a little better. However, the authors are still implying that “chronic treatment” is different from “acute treatment” by using the phrase “starting at lower concentrations – without overtly saying lower to what? Clearly that is meant to imply lower concentrations to the acute experiments in Figure 1. But there are no statistical comparisons between acute and chronic treatments – indeed the comparisons could not be direct. Your ANOVA in Figure 1 is using more conditions than the ANOVA in Figure 2 and may be affecting the significance. Additionally (see points 3-4 above), I am still unclear that the DMSO concentrations are equivalent for both acute and chronic treatments. I remain suspicious that there is no statistical (let alone biological) effect of chronic treatment in comparison to acute treatment.

6 - 9. That’s fine.

10. The line for acute 0.01 mM docetaxel in Figure 2 should be removed as it is meaningless in the context of the chronic experiment. And indeed the acute 0.01 mM docetaxel from Figure 1A is clearly under 60sec whereas the line in Figure 2A is over 60 sec – so I don’t know where this number is actually generated from?

11. That’s fine.

Reviewer #2: The authors have substantially improved the methods transparency e.g. the number of tubes/condition and the blinding procedures, while also correcting results/legend inconsistency.

**Do you want your identity to be public for this peer review?** For information about this choice, including consent withdrawal, please see our Privacy Policy

Reviewer #1: No

Reviewer #2: No

---

## [Author Response · Author response to Decision Letter 2]

19 Jan 2026

Dear Reviewers,

We want to thank the reviewers for their continuous support and constructive feedback. We have thoroughly considered each comment and revised the manuscript accordingly. We believe that the added clarifications and details have continued to strengthen the manuscript. Below, we provide our detailed responses to each point.

Reviewer 1:

Major Comments:

Acute exposure

1. That’s fine.

2. Essentially that is fine – and I am certainly not going to require you to redo your controls in light of the provided data. But I would suggest in future always doing your controls in the equivalent volume of solvent. I note in the figure provided that the mean is increasing in 0.1% DMSO (from ~35 to 45 sec) with fairly large error bars. And it may well be that the solvent is a contributing factor to the statistical effect seen in this manuscript.

Thank you for your feedback on this matter and we appreciate that you accept the supporting data we have presented. After revisiting our data and the data previously collected in our lab, we understand the importance of reporting controls that are equivalent to the volume of the solvent used. This methodology indeed clarifies any discrepancies that may arise in future reported data. For future investigations, we will ensure to include controls accordingly.

Chronic exposure

3. & 4. I am still unclear on the final concentrations for the chronic exposure experiments and think this needs to be clarified explicitly in the text. From the revised wording, the drugs are dissolved in 1% DMSO and then diluted 1/10 in M9 (equivalent to the acute experiments). For the chronic experiments, this is further dissolved in 10 mls NGM (+ 2 mls solution = 12 mls total). The wording certainly implies that the final DMSO concentrations in the chronic experiments will not be 0.1% - they will be ~0.01%. And it should also be clarified that the final drug concentrations are taking the additional volumes into account as well.

Thank you for pointing this out. After studying our calculations, we have revised our manuscript to include these specifications in lines 110, and 114-117. We acknowledge that the different DMSO concentrations used for acute versus chronic experiments do not generate a direct comparison between both models.

5. This is worded a little better. However, the authors are still implying that “chronic treatment” is different from “acute treatment” by using the phrase “starting at lower concentrations – without overtly saying lower to what? Clearly that is meant to imply lower concentrations to the acute experiments in Figure 1. But there are no statistical comparisons between acute and chronic treatments – indeed the comparisons could not be direct. Your ANOVA in Figure 1 is using more conditions than the ANOVA in Figure 2 and may be affecting the significance. Additionally (see points 3-4 above), I am still unclear that the DMSO concentrations are equivalent for both acute and chronic treatments. I remain suspicious that there is no statistical (let alone biological) effect of chronic treatment in comparison to acute treatment.

In our manuscript, we clarify that differences in plate preparation resulted in different DMSO concentrations between chronic and acute exposure experiments (Lines 110, 114-117), making these two methods not directly comparable, as no direct statistical comparison was performed. Additionally, we have changed the description title for Fig 2 to indicate that “Chronic exposure to docetaxel increased time to recovery from shock-induced seizure-like behaviors [Fig 2A] and % NR [Fig 2B], in contrast to nematodes exposed to M9 saline, which did not exhibit such changes” (Lines 205-210 & 332-333). In our Supplemental Figures document titled “S1 Appendix” we have also included this correction (Lines 403-408).

6 - 9. That’s fine.

10. The line for acute 0.01 mM docetaxel in Figure 2 should be removed as it is meaningless in the context of the chronic experiment. And indeed the acute 0.01 mM docetaxel from Figure 1A is clearly under 60sec whereas the line in Figure 2A is over 60 sec – so I don’t know where this number is actually generated from?

We have removed the line for acute 0.01 mM docetaxel in Figure 2. This change can be observed in the Figures document submitted with this revision.

11. That’s fine.

Reviewer #2: The authors have substantially improved the methods transparency e.g. the number of tubes/condition and the blinding procedures, while also correcting results/legend inconsistency.

---

## [Decision Letter · Decision Letter 2]

21 Jan 2026

Anticonvulsant effects of novel and repurposed drugs on docetaxel-induced neuropathy in C. elegans

PONE-D-25-51247R2

Dear Dr. Gonzalez-Lerma,

We’re pleased to inform you that your manuscript has been judged scientifically suitable for publication and will be formally accepted for publication once it meets all outstanding technical requirements.

Kind regards,

Michael Massiah

Academic Editor

PLOS One

Additional Editor Comments (optional):

Reviewers' comments:

Reviewer's Responses to Questions

**Comments to the Author**

Reviewer #1: All comments have been addressed

2. Is the manuscript technically sound, and do the data support the conclusions?

Reviewer #1: Yes

3. Has the statistical analysis been performed appropriately and rigorously?

Reviewer #1: Yes

4. Have the authors made all data underlying the findings in their manuscript fully available?

Reviewer #1: Yes

5. Is the manuscript presented in an intelligible fashion and written in standard English?

Reviewer #1: Yes

Reviewer #1: (No Response)

**Do you want your identity to be public for this peer review?** For information about this choice, including consent withdrawal, please see our Privacy Policy

Reviewer #1: No

---

## [Editor Report · Acceptance letter]

PONE-D-25-51247R2

PLOS One

Dear Dr. Gonzalez-Lerma,

I'm pleased to inform you that your manuscript has been deemed suitable for publication in PLOS One. Congratulations! Your manuscript is now being handed over to our production team.

Kind regards,

on behalf of

Dr. Michael Massiah

Academic Editor

PLOS One